# A dual-threshold system relying on multiple c-di-GMP metabolic enzymes controls cell fate of a cyanobacterium

Qing-Xue Sun[1], Min Huang[1¤], Yiling Yang[1], Xiaoli Zeng[1*], Cheng-Cai Zhang[1,2,3*]

**1** Key Laboratory of Algal Biology, Institute of Hydrobiology, Chinese Academy of Sciences, Wuhan, Hubei, People's Republic of China, **2** State Key Laboratory of Lake and Watershed Science for Water Security, Chinese Academy of Sciences, Nanjing, People's Republic of China, **3** Hubei Hongshan Laboratory, Wuhan, People's Republic of China

¤ Current address: Key Laboratory of Quantitative Engineering Biology, Shenzhen Institute of Synthetic Biology, Shenzhen Institute of Advanced Technology, Chinese Academy of Sciences, Shenzhen, People's Republic of China

* zengxl@ihb.ac.cn (XZ); cczhang@ihb.ac.cn (C-CZ)

## Abstract

Cyclic-di-GMP (c-di-GMP) is a ubiquitous second messenger in bacteria and regulates a variety of cell activities. Many bacteria contain multiple enzymes involved in c-di-GMP synthesis or degradation; however, how they coordinate with each other to orchestrate c-di-GMP homeostasis remains unclear. Here, using the cyanobacterium *Anabaena* PCC 7120 as a model, we created $cdG^0$ and $cdG^{max}$ strains by deleting all 8 and 14 genes, respectively, that encode enzymes with c-di-GMP degradation and synthesis domains, alongside a collection of mutants with various numbers of these genes deleted. Our findings demonstrate that c-di-GMP in *Anabaena* not only modulates cell size but is also indispensable for cell viability. Quantitative analysis established two critical physiological thresholds in vivo: a minimal c-di-GMP level required for cell size maintenance and a lower, lethal threshold essential for survival. We show that the 16 enzymes involved in c-di-GMP turnover in *Anabaena* function as an electromechanical-like dual relay to control c-di-GMP dynamics, with different modules contributing to c-di-GMP homeostasis or responding as an SOS alarm when the c-di-GMP concentration drops below the lethal threshold. Both effects of c-di-GMP on cell size reduction and cell viability are mediated by the cyclic-di-GMP receptor (CdgR), depending on the amount of the c-di-GMP-free form of CdgR available because of titration by c-di-GMP in the cells. The system, with the two concentration thresholds of c-di-GMP that dictate cell size and viability, respectively, enables dynamic cellular adaptation while preventing lethal effects.

**Data availability statement:** Data are available in the published article and its online supplemental material. All numerical data are provided in S1 Data, which contains multiple datasheets. All raw images are provided in S1 Raw Images, and the raw microscopy images have been deposited in Figshare (https://doi.org/10.6084/m9.figshare.31842808). Plasmid pCpf1b-sp was deposited at Addgene with ID number #122188. Plasmid pCT was submitted to GenBank with ID number MK948095.

**Funding:** This research was supported by the Youth Innovation Promotion Association CAS (Grant No. 2022342, to X.Z., http://yicas.cn/about/jianjie/), the National Natural Science Foundation of China (Grant No. 32270063, to X.Z., https://www.nsfc.gov.cn/), the Strategic Priority Research Program of the Chinese Academy of Sciences (Grant No. XDB0480000, to C-C.Z, https://www.cas.cn/), and the China Postdoctoral Science Foundation (Grant No. 2023M743726, to Q-X.S, https://www.chinapostdoctor.org.cn/bshjjh#). Funders do not play any role in the study design, data collection and analysis, decision to publish, or preparation of the manuscript.

**Competing interests:** The authors have declared that no competing interests exist.

**Abbreviations:** 8 PDEs, 8 c-di-GMP hydrolases; CBB, Coomassie Brilliant Blue; c-di-GMP, cyclic-di-GMP; CdgR, c-di-GMP receptor; DGCs, diguanylate cyclases; LC-MS, liquid chromatography-mass spectrometry; PAS, Per–Arnt–Sim; PDEs, phosphodiesterases; qRT-PCR, quantitative real-time PCR; SEC, size-exclusion chromatography; WT, wild-type.

## Introduction

cyclic-di-GMP (c-di-GMP), a bacterial second messenger, governs a large spectrum of cellular processes [1–5]. The regulatory effects of c-di-GMP are mediated by binding to specific receptors, which exhibit remarkable structural and sequence diversity across bacterial species [6–9]. These regulatory effects can occur at multiple levels, including transcriptional, post-transcriptional, and post-translational regulation [2,10–13].

The intracellular levels of c-di-GMP are finely regulated by two classes of enzymes: diguanylate cyclases (DGCs) and phosphodiesterases (PDEs) [2]. DGCs, characterized by a conserved GGDEF domain, are responsible for synthesizing c-di-GMP by the condensation of two GTP molecules [2,14]. PDEs are responsible for the degradation of c-di-GMP. PDEs typically possess either an EAL domain or an HD-GYP domain, and hydrolyze c-di-GMP into linear pGpG or eventually into GMP [2,15]. Importantly, GGDEF, EAL, and HD-GYP domains are often found within the same protein, and their output activities can be regulated by various sensory domains that respond to a range of environmental signals [4,5,16,17]. Additionally, DGCs often possess a c-di-GMP-binding inhibitory (I)-site involved in allosteric feedback inhibition [18]. Consequently, the intracellular levels of c-di-GMP can be finely tuned in response to internal and external environmental cues.

Most bacterial genomes contain multiple genes related to c-di-GMP turnover [16]. For example, *Escherichia coli* and *Pseudomonas aeruginosa* contain 29 and 41 such genes, respectively [16]. This genetic abundance implies sophisticated regulation mechanisms to control the c-di-GMP pools in response to various environmental cues. Some studies have revealed critical insights into c-di-GMP-mediated regulation [19–23]. For example, in *Salmonella*, most GGDEF proteins are constitutively and independently expressed [20]. In *Caulobacter crescentus*, c-di-GMP controls multiple developmental pathways, each with a distinct activation concentration [21]. Additionally, in *Dickeya zeae* EC, 3 out of 19 c-di-GMP turnover proteins play dominant roles in modulating the global c-di-GMP pool [23]. These findings collectively indicate the complexity and adaptability of c-di-GMP signaling networks in bacterial systems. However, it remains unclear why bacteria possess so many genes for c-di-GMP turnover and what the functional purpose and interplay of these genes are.

*Anabaena* sp. PCC 7120 (*Anabaena*) is a filamentous cyanobacterium that houses 16 genes for c-di-GMP metabolism [16,24,25]. We recently identified c-di-GMP as an intracellular proxy for cell size control [26] and discovered a highly conserved c-di-GMP receptor (CdgR), which modulates cell size through interaction with DevH [7,27]. While individual deletion of the 16 genes led to the identification of CdgS as the primary DGC responsible for cell size regulation, by forming a two-component system with the histidine kinase CdgK, other single mutants had little noticeable phenotype [24,26,28]. Therefore, potential compensatory or redundant functions of the remaining 15 c-di-GMP metabolic proteins in cell size control and other physiological processes need to be elucidated.

To investigate how c-di-GMP turnover proteins modulate physiology in *Anabaena*, we methodically engineered two mutants: a c-di-GMP-deficient strain (*cdG⁰*) and

a c-di-GMP-overproducing strain (*cdG^max*) by sequentially deleting all 14 GGDEF-domain genes and all 8 EAL/HD-GYP domain genes, respectively. The deletion of such a large number of genes is unprecedented in cyanobacterial genetics and allowed us to determine the physiological functions of c-di-GMP. Several mutants that produce intermediate levels of c-di-GMP in this process were also obtained. By systematic analysis, we identified the dominant DGCs and PDEs in this cyanobacterium. We propose that the c-di-GMP metabolic enzymes function in a manner that simulates an electromechanical dual relay system to control the c-di-GMP homeostasis. In this system, 13 enzymes constitute a baseline signal pool, two acting as a responsive module, and one as an emergency module that is activated only when c-di-GMP concentration drops to a lethal level. These functional modules determine two concentration thresholds that dictate cell size and viability, respectively.

## Results

### Analysis of a mutant in which all 8 genes encoding c-di-GMP hydrolase were deleted demonstrates that high c-di-GMP levels do not affect cell growth and cell size

*Anabaena* possesses 16 proteins involved in c-di-GMP turnover [16,26]. These include 8 proteins with GGDEF domains (DGC-only), 2 proteins containing EAL/HD-GYP domains (PDE-only), and 6 hybrid proteins that contain both types of protein domains (Dual-function) (Fig 1A). Previous studies revealed that neither deleting a single protein containing the EAL domain nor overexpressing the heterologous DGC YdeH had any observable impact on cell size or cell growth [24,26]. This suggested the functional redundancy among the 8 c-di-GMP hydrolases (8 PDEs).

To overcome this redundancy and investigate the impact of high c-di-GMP levels in *Anabaena*, we systematically deleted all 8 PDE genes in frame based on CRISPR-Cpf1 [29], generating a PDE-null mutant (*8ΔPDE, cdG^max*) (Figs 1B and S1) and seven intermediate strains (*1ΔPDE, 2ΔPDE, 3ΔPDE, 4ΔPDE, 5ΔPDE, 6ΔPDE, 7ΔPDE* (Fig 1B). Liquid chromatography-mass spectrometry (LC-MS) quantification of intracellular c-di-GMP revealed that a slight increase was observed upon deletion of the first single gene, *alr2306* (*1ΔPDE*), and the levels in the other 5 mutants (*2ΔPDE, 3ΔPDE, 4ΔPDE, 5ΔPDE, 6ΔPDE*) showed no significant difference from the WT strain (Fig 2A). Deletion of the 7th PDE gene (*all4897*) triggered a sharp increase in c-di-GMP, and this level was further elevated in the *cdG^max* strain, reaching a peak approximately 3.6-fold higher than WT following the deletion of *alr3170* (Fig 2A). In contrast, Δ*all3170* and Δ*all4897* single mutants exhibited WT c-di-GMP levels (Fig 2A), suggesting functional redundancy and cross regulation among these PDE enzymes. Next, we analyzed the cell morphology of all mutants and found that they all displayed cell length and width similar to the WT strain (Figs 2B and S2). The *cdG^max* strain also showed no growth defects under either BG11 or BG110 conditions, compared to the WT (S3 Fig). Together, these findings demonstrate that elevated c-di-GMP levels do not affect cell growth and cell size in *Anabaena*.

### c-di-GMP is essential for cell survival

Previous studies demonstrated that the deletion of *cdgS* (strain Δ*cdgS* or *1ΔDGC*) significantly reduced cell size [26,28]. Quantification analysis revealed that the intracellular c-di-GMP level was ~30 ± 11% lower than that in the WT strain (Fig 2C). To explore the effects of additional c-di-GMP reduction and the roles of other DGCs in c-di-GMP homeostasis, we attempted to construct a DGC-null mutant (*cdG^0*) by systematically deleting all 14 genes encoding GGDEF domain-containing proteins (Fig 1B). While 13 of these genes were successfully deleted (*13ΔDGC*), the final gene, *all1219*, could not be inactivated in this background, while *all1219* alone could be inactivated as reported by two independent studies [26,28]. This suggests that c-di-GMP may be essential for the survival of *Anabaena*.

To overcome this challenge, we generated a conditional mutant by replacing the native promoter of *all1219* with the $Cu^{2+}$ and theophylline (TP) inducible one (*CT* promoter) [24] in the *13ΔDGC* background strain, generating the *14ΔDGC* (*cdG^0*) strain (S4 Fig). When cultured in BG11 medium supplemented with 0.5 mM TP and 0.3 μM $Cu^{2+}$, the *cdG^0* strain

**A**

| Class | Gene ID | Functional Conclusions from This Study | Domain architectures | GGDEF motif | EAL motif | RxxD motif |
|---|---|---|---|---|---|---|
| DGC-only | all1012 | | REC PAS GGDEF | GGDEF | — | RSGD |
| | all1219 | major DGC | CHASE 2 PAS GGDEF | GGDEF | — | RASD |
| | all2416 | | GGDEF | GGEEF | — | RSQD |
| | all2874 | major DGC (CdgS) | REC GGDEF | GGEEF | — | RPAD |
| | alr3504 | | GGDEF | GGDEF | — | RKID |
| | alr3599 | emergency DGC | REC GGDEF | GGEEF | — | KHQD |
| | all4896 | | PAS GAF GGDEF | GGDEF | — | RSFD |
| | all5174 | | REC Trans regc HPT GGDEF | GGTEF | — | HSQD |
| Dual-function | all0219 | | PAS GGDEF EAL | GGDEF | EAL | RPTD |
| | all1175 | | FHA PAS GAF GGDEF EAL | GGDEF | EAL | RSGD |
| | alr2306 | major PDE | REC GGDEF EAL | GGDEF | EAL | TPDA |
| | alr3170 | | PAS PAS PAS GGDEF EAL | GGDEF | EAL | QPED |
| | all4225 | | GGDEF EAL | SGDEG | EAL | AQKD |
| | all4897 | | PAS GGDEF EAL | GGDEF | EAL | RAGD |
| PDE-only | alr1230 | | REC EAL | — | ESL | — |
| | alr3920 | | REC HDc | — | HDIG | — |

**B**

| Strain | Genotype |
|---|---|
| 1ΔPDE | Δalr2306 |
| 2ΔPDE | Δalr2306 Δall0219 |
| 3ΔPDE | Δalr2306 Δall0219 Δall4225 |
| 4ΔPDE | Δalr2306 Δall0219 Δall4225Δalr1230 |
| 5ΔPDE | Δalr2306 Δall0219 Δall4225 Δalr1230 Δalr3920 |
| 6ΔPDE | Δalr2306 Δall0219 Δall4225 Δalr1230 Δalr3920 Δall1175 |
| 7ΔPDE | Δalr2306 Δall0219 Δall4225 Δalr1230 Δalr3920 Δall1175 Δall4897 |
| 8ΔPDE (cdG$^{max}$) | Δalr2306 Δall0219 Δall4225 Δalr1230 Δalr3920 Δall1175 Δall4897 Δalr3170 |
| 1ΔDGC | ΔcdgS |
| 2ΔDGC | ΔcdgSΔalr3504 |
| 3ΔDGC | ΔcdgSΔalr3504Δalr3599 |
| 4ΔDGC | ΔcdgSΔalr3504Δalr3599Δall2416 |
| 5ΔDGC | ΔcdgSΔalr3504Δalr3599Δall2416Δall1012 |
| 6ΔDGC | ΔcdgSΔalr3504Δalr3599Δall2416Δall1012Δall5174 |
| 7ΔDGC | ΔcdgSΔalr3504Δalr3599Δall2416Δall1012Δall4896Δall4897 |
| 8ΔDGC | ΔcdgSΔalr3504Δalr3599Δall2416Δall1012Δall4896Δall4897Δalr3170 |
| 9ΔDGC | ΔcdgSΔalr3504Δalr3599Δall2416Δall1012Δall4896Δall4897Δalr3170Δall0219 |
| 10ΔDGC | ΔcdgSΔalr3504Δalr3599Δall2416Δall1012Δall4896Δall4897Δalr3170Δall0219Δall4225 |
| 11ΔDGC | ΔcdgSΔalr3504Δalr3599Δall2416Δall1012Δall4896Δall4897Δalr3170Δall0219Δall4225Δall1175 |
| 12ΔDGC | ΔcdgSΔalr3504Δalr3599Δall2416Δall1012Δall4896Δall4897Δalr3170Δall0219Δall4225Δall1175Δall5174 |
| 13ΔDGC | ΔcdgSΔalr3504Δalr3599Δall2416Δall1012Δall4896Δall4897Δalr3170Δall0219Δall4225Δall1175Δall5174Δalr2306 |
| 14ΔDGC(cdG$^0$) | ΔcdgSΔalr3504Δalr3599Δall2416Δall1012Δall4896Δall4897Δalr3170Δall0219Δall4225Δall1175Δall5174Δalr2306Δall1219 |

**Fig 1. Enzymes related to c-di-GMP turnover in *Anabaena* and their corresponding mutants. (A)** Predicted domain architectures and consensus sequence motifs of 16 proteins for c-di-GMP synthesis and degradation. Abbreviations: GGDEF, diguanylate cyclase (DGC) domain; EAL, phosphodiesterase (PDE) domain; REC: receiver domain; PAS: Per–ARNT–Sim domain; GAF: cGMP phosphodiesterase/adenylate cyclase/FhlA domain; FHA:

fork head-associated domain; orange rectangle: transmembrane domain. **(B)** Deletion mutants were generated in this study. The $cdG^0$ strain (*14ΔDGC*) is a conditional mutant in which 13 genes were deleted, while for the 14th one (*all1219*), the promoter region was replaced by a $Cu^{2+}$ and theophylline inducible platform (CT) at the native chromosomal locus.

exhibited cell growth (Fig 2D) and cell size (Fig 2E and 2F) comparable to those of the WT strain, with c-di-GMP levels approximately 1.4-fold higher than WT (Fig 2C). Upon inducer removal, the intracellular c-di-GMP level gradually decreases, reaching an undetectable level by 72 h (Fig 2C). Subsequently, by 120 h post-inducer removal, $cdG^0$ strain exhibited a complete growth arrest (Figs 2D and S4C). These results demonstrate that the CT promoter could tightly regulate *all1219* expression, and confirm that c-di-GMP is essential for cell viability in *Anabaena*.

Consistent with previous reports [26], under conditions of c-di-GMP depletion (achieved by the removal of $Cu^{2+}$ and TP), the average cell length and width of the $cdG^0$ strain gradually decreased over time (Fig 2E and 2F). At 48 h after removing inducer, as the intracellular c-di-GMP dropped to $55 \pm 11\%$ of that found in the WT strain, the average cell length and width of the $cdG^0$ strain were reduced to $2.6 \pm 0.32$ μm and $3.0 \pm 0.19$ μm, respectively, compared to $4.1 \pm 0.54$ μm and $3.6 \pm 0.16$ μm in WT strain (Fig 2E and 2F). A slight recovery in the average cell length and width was observed at 72 h and 96 h (Fig 2E and 2F), even though the c-di-GMP level had become undetectable (Fig 2B). However, this recovery coincided with filament fragmentation and a gradual arrest in cell growth (Fig 2E and 2F), suggesting the existence of a transient compensatory mechanism that ultimately fails to sustain normal cellular function.

To further confirm that the c-di-GMP reduction is responsible for the defects of the $cdG^0$ strain, we overexpressed a well-characterized heterologous DGC, YdeH, or its inactive variant $YdeH^{GGAAF}$ [24,30] in this background, generating the strains $cdG^0$::*ydeH* and $cdG^0$::*ydeH*$^{GGAAF}$, respectively. Upon the inducer removal, the expression of YdeH in the $cdG^0$::*ydeH* strain resulted in the production of c-di-GMP levels approximately 5.6-fold higher than those in the WT strain, and it fully restored the cell size and cell growth defect of the $cdG^0$ strain (Fig 2G and 2H). In contrast, the inactive YdeH-$^{GGAAF}$ was unable to restore either the phenotype or c-di-GMP level (Fig 2G and 2H). These results indicated that the c-di-GMP level is directly responsible for the defects in cell size and cell viability observed in the $cdG^0$ strain and that higher c-di-GMP levels did not result in a detectable phenotype, consistent with the results obtained in the $cdG^{max}$ mutant (Fig 2A and 2B). Collectively, these findings indicate that c-di-GMP is essential for maintaining cell size and sustaining cell growth in *Anabaena*.

## Alr2306 is the major PDE to maintain intracellular c-di-GMP for cell growth

During the process of constructing a DGC-null mutant, 12 intermediate mutants with different numbers of genes deleted (*1ΔDGC, 2ΔDGC, 3ΔDGC, 4ΔDGC, 5ΔDGC, 7ΔDGC, 8ΔDGC, 9ΔDGC, 10ΔDGC, 11ΔDGC, 12ΔDGC, 13ΔDGC*) were generated. Microscopic observations revealed that, in contrast to the other 11 mutants, which exhibited a significant reduction in both cell length and cell width, the *13ΔDGC* mutant recovered cell length and width similar to the WT strain (Figs 3A and S5). Quantitative analysis further demonstrated that the intracellular c-di-GMP level in the *13ΔDGC* strain was restored to $88 \pm 3.7\%$ of the WT level, whereas the other mutants exhibited c-di-GMP levels ranging from $53 \pm 0.79\%$ to $75 \pm 6.2\%$ of the WT level (Fig 3B). Since the 13th gene deleted in the *13ΔDGC* strain was *alr2306*, and the deletion of *alr2306* alone (*1ΔPDE*) caused a slight increase in c-di-GMP levels (Fig 2A), we proposed that Alr2306, which contains a receiver (REC) domain at the N-terminal, a GGDEF domain, and an EAL domain at the C-terminal (Fig 1A), may play a critical role as a major PDE in regulating cell size and growth by modulating c-di-GMP levels.

To validate this hypothesis, we constructed a series of double mutants by deleting other c-di-GMP metabolic genes in the background of the Δ*cdgS* (*1ΔDGC*) strain, which has been previously identified as the primary DGC involved in cell size regulation [26]. Except for the Δ*cdgS*Δ*all1219* double mutant, we successfully generated 14 other double mutants (Figs 3C and S5), highlighting the potential importance of *all1219* in c-di-GMP synthesis, which is described

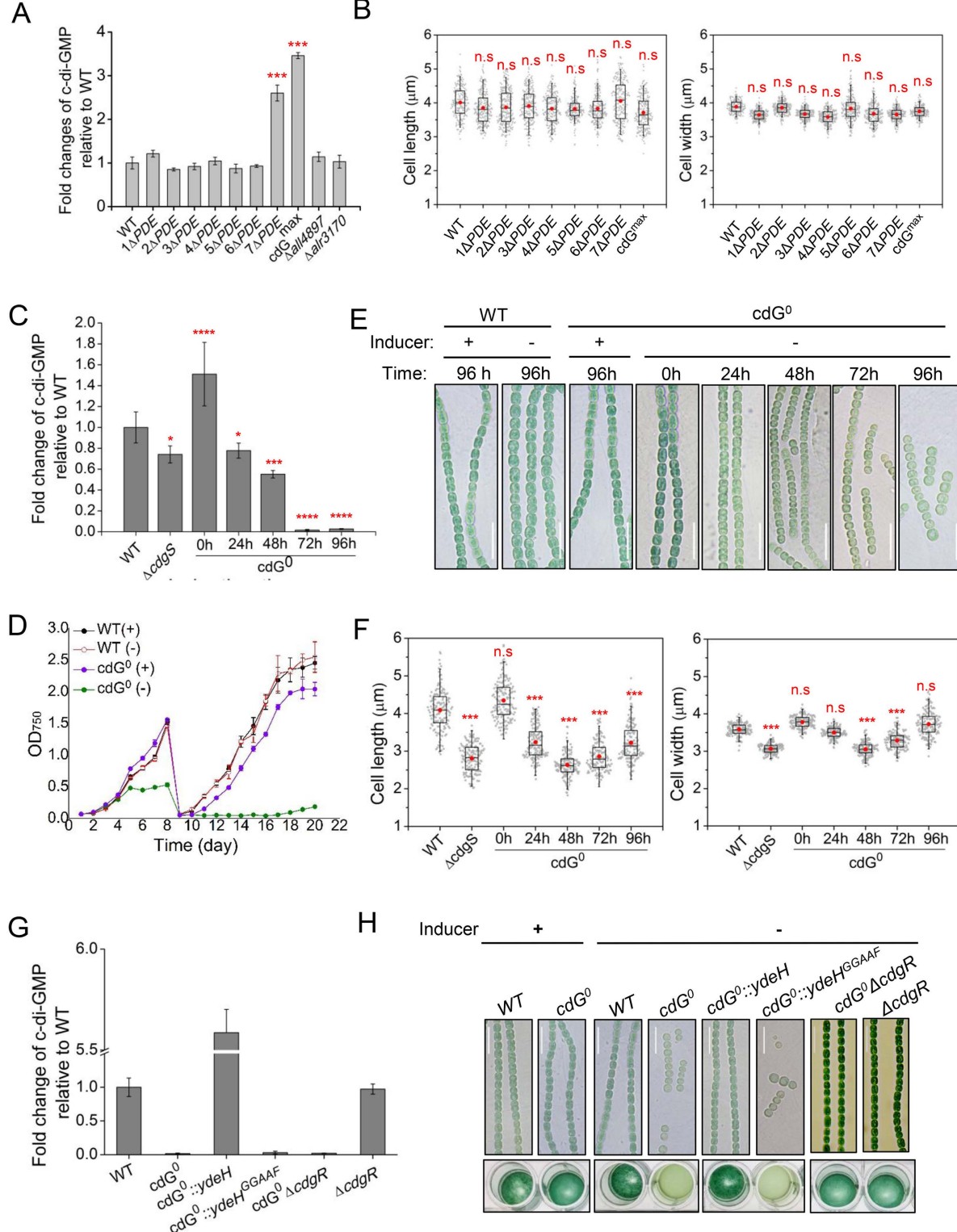

**Fig 2. The phenotype and c-di-GMP level of all PDE deletion strains and the *cdG⁰* strain. (A)** The intracellular c-di-GMP levels of the indicated strains. **(B)** Statistical analysis of cell length and cell width of the indicated strains based on images, as shown in S2 Fig, using a box plot. **(C)** The intracellular c-di-GMP levels of the *cdG⁰* strain during the time course of $Cu^{2+}$ and theophylline depletion. **(D)** The growth curves of the *cdG⁰* strain and WT strain in BG11 medium with (+) or without (−) $Cu^{2+}$ and theophylline. Absorbance at 750 nm was measured at the indicated time points. To better remove

traces of $Cu^{2+}$ and theophylline, the cultures were re-inoculated for the second time in a medium free of $Cu^{2+}$ and theophylline after 8 days of incubation. All values are shown as mean ± standard deviation, calculated from triplicate data. **(E)** Micrographs of *Anabaena* filaments of WT and *cdG⁰* strains cultured in BG11 medium with (+) or without (−) $Cu^{2+}$ and theophylline at the indicated time points. Scale bars represent 15 μm. **(F)** Statistical analysis of cell size parameters of the *cdG⁰* strain during the time course of $Cu^{2+}$ and theophylline depletion. The cell length and cell width were measured based on images, as shown in (E). **(G)** The intracellular c-di-GMP level of the indicated strains at 96 h in BG11 medium without $Cu^{2+}$ and theophylline. **(H)** Top panel: the micrographs of the indicated strains cultured in BG11 medium with (+) or without (−) $Cu^{2+}$ and theophylline at 96 h. Bottom panel: the growth of different strains, as in the top panel tested in 24-well plates. All cultures started with a similar OD at 0.3 diluted from a pre-culture and imaged after 4 days of incubation. In (A), (C), and (G), the c-di-GMP concentrations of all mutants were normalized to the levels of the wild-type (WT) strain. The fold change numbers are shown as mean ± SD from three biological replicates. In (B) and (F), 150–200 cells of each strain from three independent experiments were measured. The boxplots enclose the 25th and the 75th percentile, with the black line representing the median value, red dot representing the mean value. The statistical significance in comparison to WT was carried out by a two-sided Student *t* test. The red asterisks indicate significance in comparison to WT as follows: *$p < 0.05$; **$p < 0.01$; ***$p < 0.001$. ****$p < 0.0001$, n.s.: not significant ($p > 0.05$). WT, wild-type *Anabaena*. The data underlying this Figure can be found in S1 Data. The raw images underlying this Figure can be found in S1 Raw Images.

later in this study. Among the 14 double mutants, only the deletion of *alr2306* suppressed the reduced cell length and width caused by the deletion of *cdgS* (Figs 3C and S5). Consistent with this observation, the intracellular c-di-GMP level of the Δ*cdgS*Δ*all2306* double mutant was also restored to the WT level (Fig 3D). As a control, the complemented strain Δ*cdgS*Δ*alr2306::alr2306* reverted to cell size and c-di-GMP level back to those observed in Δ*cdgS* (Figs 3C, 3D, and S5). These results demonstrate that the deletion of *alr2306* can elevate the intracellular c-di-GMP levels in *Anabaena* and could partially compensate for the reduced c-di-GMP levels resulting from the loss of *cdgS*. Therefore, although Alr2306 has both a c-di-GMP synthesis domain and a degradation domain, it functions as one of the major PDEs responsible for controlling intracellular c-di-GMP levels, thereby regulating cell size and growth.

**CdgS, Alr1219, and Alr3599 are the major DGCs that maintain intracellular c-di-GMP levels for cell survival**

Because of the initial failure to obtain the Δ*cdgS*Δ*all1219* double mutant, we hypothesized that Alr1219, which contains a CHASE (cyclase and histidine kinase associated sensory extracellular) domain, a transmembrane domain, a PAS (Per–Arnt–Sim) domain, and a GGDEF domain (Fig 1A), functions together with CdgS as the major DGCs for maintaining cell survival of *Anabaena*. To test this, we generated a conditional strain, Δ*cdgSCT-all1219*, in which the expression of *all1219* is controlled by the CT promoter, as described above. Under induction (0.5 mM TP and 0.3 μM $Cu^{2+}$), the Δ*cdgSCT-all1219* exhibited cell growth and cell size comparable to those of the WT strain, with intracellular c-di-GMP levels ~1.2-fold higher than the WT (Fig 4A–4C). Surprisingly, upon inducer removal, the Δ*cdgSCT-all1219* strain was still able to grow similarly to the WT strain (Fig 4A and 4E). Microscopic observations revealed that the cell size of the Δ*cdgSCT-all1219* strain initially decreased significantly, but started to recover at 72 h, even back to a cell size similar to the WT strain by 96 h (Fig 4C). Further quantitative analysis revealed that, after the removal of $Cu^{2+}$ and TP, the intracellular c-di-GMP levels in the Δ*cdgSCT-all1219* strain first decreased dramatically. By 48 h, the intracellular c-di-GMP level dropped to about 42 ± 0.2% of that in the WT strain. However, after 48 h, the intracellular c-di-GMP level began to recover, eventually reaching up to 4-fold higher than that in the WT strain by 96 h (Fig 4B). The restoration of c-di-GMP levels explains the growth capacity observed in the Δ*cdgSCT-all1219* strain after the inducer removal and suggests that some additional c-di-GMP turnover enzymes may have been activated under conditions when c-di-GMP levels become severely limited.

To identify these enzymes responsible, we constructed a new mutant Δ*8DGC-1,* in which all 8 genes encoding DGC enzymes alone were deleted (Δ*cdgS*Δ*alr3504*Δ*alr3599* Δ*all2416*Δ*all1012*Δ*all5174*Δ*all4896CT-all1219*), while the expression of *all1219* is controlled by the CT promoter. Upon removal of the inducer, this Δ*8DGC-1* strain failed to grow, exhibited a significant reduction in cell size, and finally entered growth arrest (Fig 4A and 4D). These phenotypic observations were supported by intracellular c-di-GMP measurements: following $Cu^{2+}$ and TP depletion, c-di-GMP levels in the Δ*8DGC-1* strain decreased rapidly, becoming undetectable by 48 h (Fig 4B). Although a partial recovery of c-di-GMP levels was observed after 72 h, reaching ~40 ± 0.7% of WT levels by 120 h, this c-di-GMP level was insufficient to support cell growth

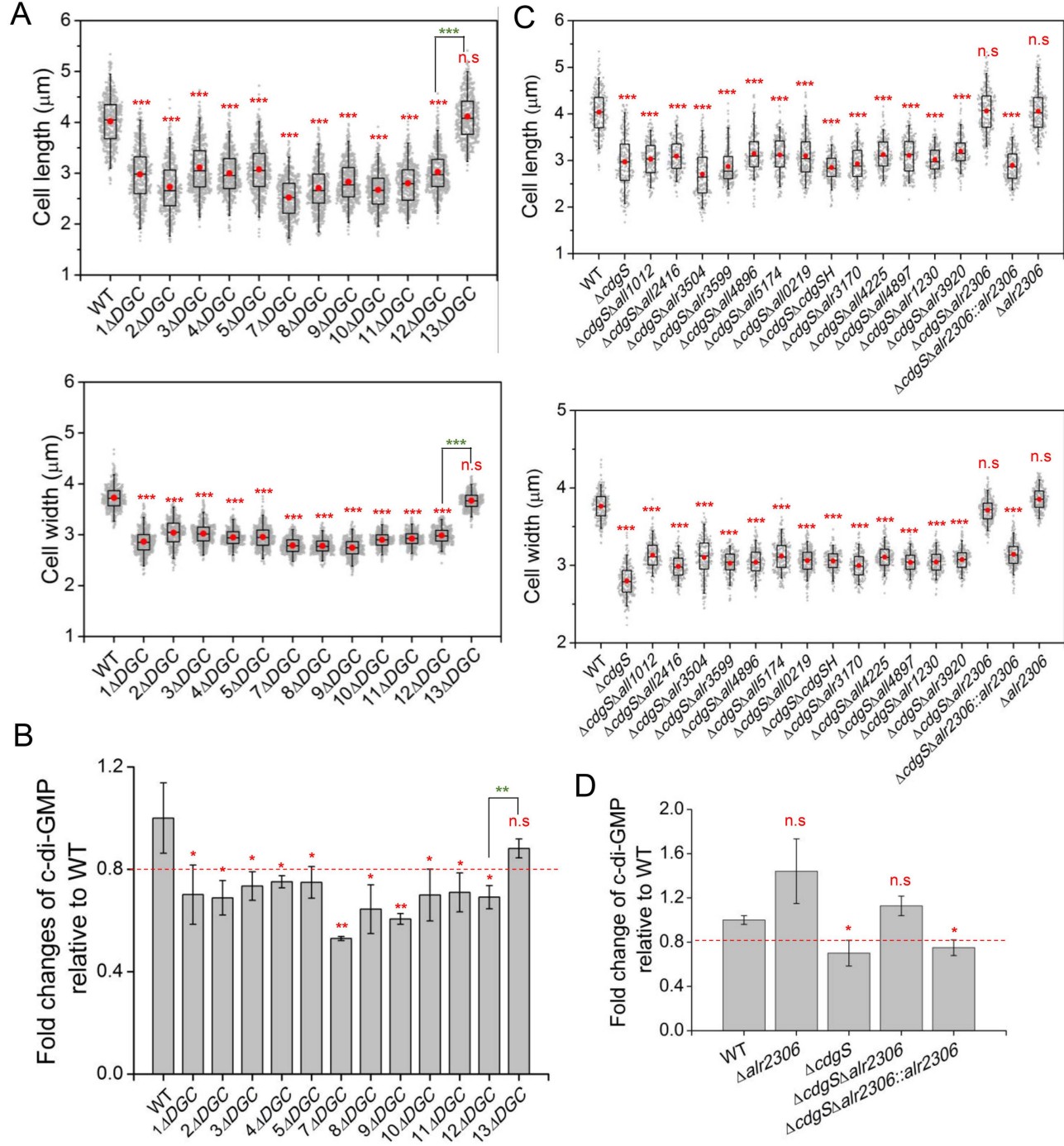

**Fig 3. Alr2306 is the major PDE for cell size maintenance. (A and C)** Statistical analysis of cell size parameters of the indicated strains. The cell length (upper panel) and cell width (lower panel) were measured based on images, as shown in S4 Fig 200 cells of each strain from three independent experiments were measured. The boxplots enclose the 25th and the 75th percentile, with the black line representing the median value, red dot representing the mean value. **(B and D)** The intracellular c-di-GMP level of the indicated strains. The c-di-GMP concentrations of all mutants were normalized to the levels of the WT strain. The fold change numbers are shown as mean±SD from three biological replicates. The red asterisks indicate significance in comparison to WT or between the two strains connected by a bracket at the top as follows: $*p < 0.05$; $**p < 0.01$; $***p < 0.001$. $****p < 0.0001$, n.s.: not significant ($p > 0.05$). The data underlying this Figure can be found in S1 Data. The raw images underlying this Figure can be found in S1 Raw Images.

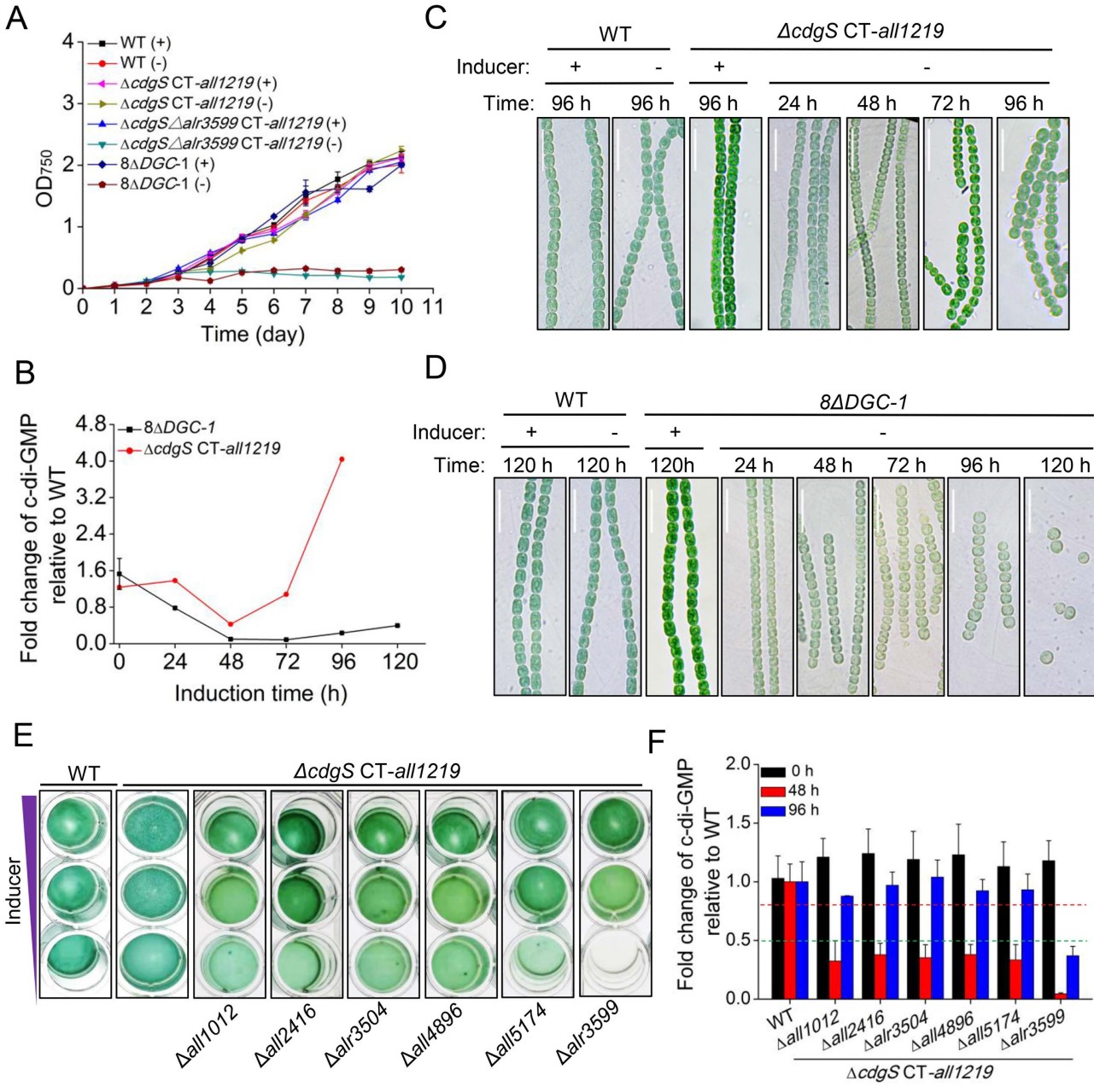

**Fig 4. The phenotype and c-di-GMP levels of *cdgS*, *alr3599*, and *alr1219*-related mutant strains. (A)** The growth curves of the indicated conditional mutant strains and the WT strain in BG11 medium with (+) or without (−) Cu$^{2+}$ and theophylline. The promoter region of *all1219* in the Δ*cdgSCT-all1219* strain, the Δ8DGC-1 strain, and the Δ*cdgSΔalr3599CT-all1219* strain was replaced by a Cu$^{2+}$ and theophylline inducible platform (CT) at the native chromosomal locus. Absorbance at 750 nm was measured at the indicated time points. All values are shown as mean ± standard deviation, calculated from triplicate data. **(B)** The intracellular c-di-GMP levels of the Δ*cdgSCT-all1219* strain and the Δ*8DGC-1* strain during the time course of Cu$^{2+}$ and theophylline depletion. **(C and D)** Micrographs of *Anabaena* filaments of WT, Δ*cdgSpICT-all1219* strains (C), and Δ*8DGC-1* strain (D) cultured in BG11 medium with (+) or without (−) Cu$^{2+}$ and theophylline at the indicated time points. Scale bars represent 15 μm. **(E)** The growth of indicated strains was tested in 24-well plates. All cultures started with a similar OD at 0.3 diluted from a pre-culture and imaged after 4 days of incubation. The purple triangle indicated the decreasing levels of the inducers added to the culture media (from top to bottom column): 0.3 μM copper and 2 mM theophylline, 0.125 μM copper and 0.5 mM theophylline, and 0 μM copper and 0 mM theophylline. **(F)** The intracellular c-di-GMP levels of the indicated strains at 0 h, 48 h, and 96 h of Cu$^{2+}$ and theophylline depletion. The green and red dotted line represents the 50% and 80% of the c-di-GMP level to WT, respectively. The data underlying this Figure can be found in S1 Data. The raw images underlying this Figure can be found in S1 Raw Images.

(Fig 4A and 4B). These results strongly suggest that the activated enzymes in the Δ*cdgSCT-all1219* strain responsible for c-di-GMP restoration are indeed among those absent in Δ*8DGC-1*. Furthermore, these results indicate that the intracellular c-di-GMP level at 40% ± 0.7% of WT is below the threshold required to sustain cell growth, highlighting the critical role of DGC-only enzymes in maintaining c-di-GMP homeostasis and supporting cellular proliferation.

Subsequent triple mutant screening revealed that only the Δ*cdgS*Δ*alr3599CT-all1219* strain exhibited complete cell growth arrest upon removal of Cu²⁺ and TP (Fig 4E). Quantitative analysis of the intracellular c-di-GMP levels revealed distinct patterns among these mutants (Fig 4F). The Intracellular c-di-GMP levels in the Δ*cdgS*Δ*alr3599CT-all1219* strain plummeted to around 4.6 ± 0.7% of WT levels at 48 h after inducer depletion and only partially recovered to about 37 ± 8.0% of WT levels by 96 h (Fig 4F). This limited recovery was insufficient to support cell growth, leading to growth arrest (Fig 4A and 4E). In the other five triple mutants generated, the c-di-GMP levels dropped to ~35% ± 2.5% of the WT levels at 48 h after inducer depletion, and then began to recover thereafter, eventually reaching levels close to the WT by 96 h (Fig 4F), similar to the Δ*cdgSCT-all1219* strain. This recovery suggests that, if *alr3599* is present, these mutants retain the capacity to restore c-di-GMP homeostasis for cell survival in the absence of CdgS and All1219. Taken together, we concluded that CdgS, Alr1219, and Alr3599 are the major DGCs that maintain intracellular c-di-GMP levels for cell survival, accounting for about 60% of the total c-di-GMP production under the tested conditions. Meanwhile, the limited c-di-GMP level recovery observed in the Δ*cdgS*Δ*alr3599CT-all1219* strain suggests that additional PDE-only and Dual-function enzymes contribute to c-di-GMP level maintenance, responsible for the remaining about 40% of c-di-GMP production in the absence of the CdgS, Alr1219, and Alr3599. Our results also suggest that Alr3599 functions as an emergency DGC to recover the c-di-GMP level independently, once c-di-GMP levels drop below the threshold necessary for cell viability.

### The accumulation of Alr3599, lacking auto-inhibition activity, accounts for its ability to restore c-di-GMP levels under emergency

To characterize the three major DGCs (CdgS, Alr1219, and Alr3599), the Full-length CdgS and Alr3599, along with a truncated Alr1219 variant (Alr1219-ΔCT, lacking both the N-terminal CHASE2 domain and transmembrane domain), were purified from *E. coli* and assessed for their enzymatic activities. HPLC-based activity assays revealed that all three DGCs exhibited GTP-dependent c-di-GMP synthesis activity (S6A Fig) [26]. Although all three enzymes followed Michaelis–Menten kinetics, they exhibited significantly different kinetic parameters (S6B Fig). Most notably, Alr3599 displayed an exceptionally high turnover rate ($k_{cat}$ = 4223.8 ± 1081.1 s⁻¹), ~5.3- and 9.2-fold higher than CdgS and Alr1219-ΔCT, respectively. Although Alr3599 showed comparatively lower substrate affinity ($K_m$ = 84.2 ± 30.1 μM) compared to CdgS ($K_m$ = 26.6 ± 3.5μM), and Alr1219-ΔCT ($K_m$ = 20.3 ± 12.4 μM), its superior turnover rate resulted in the highest catalytic efficiency ($k_{cat}/K_m$ = 50.2 ± 12.8 μM⁻¹ s⁻¹) (S6B Fig). In addition, a pronounced divergence in their regulatory properties was also observed. While Alr1219 and CdgS, containing canonical I-sites (RXXD) (Fig 1A), were strongly inhibited by c-di-GMP ($K_i$ = 3.7 μM and 2.42 μM, respectively), Alr3599 with its variant I-site (KXXD) showed markedly reduced sensitivity ($K_i$ = 16.0 μM) (S6C Fig). Taken together, these biochemical characteristics suggested the distinct physiological roles of the three DGCs: Alr3599's high-output, feedback-resistant design is optimized for rapid c-di-GMP production during emergency responses, while CdgS and Alr1219-ΔCT's high-affinity, tightly regulated activity maintains precise c-di-GMP homeostasis during standard growth conditions.

We next investigated the underlying regulation mechanism of Alr3599-induced activity during c-di-GMP depletion. qRT-PCR analysis showed that the transcriptional level of *alr3599* in the Δ*cdgSCT-alr1219* strain showed no significant changes following inducer depletion (S6D Fig), despite intracellular c-di-GMP levels dropping to ~42% of WT concentrations at 48 h (Fig 4B). However, western blot analysis of a Flag-tagged Alr3599 strain (Δ*cdgSCT-alr1219* (*alr3599-Flag*), S7 Fig) revealed a striking post-translational regulation pattern. Alr3599 protein levels were maintained at barely detectable levels during normal conditions (with inducer), but accumulated dramatically after inducer removal, peaking at 48 h (S6E Fig). This accumulation correlated temporally with c-di-GMP depletion as a trigger, followed by progressive

attenuation of protein levels as cellular c-di-GMP concentrations recovered (Fig 4B). These results suggest that Alr3599 controls the intracellular level of c-di-GMP at a post-transcriptional level in a c-di-GMP-dependent manner.

## Threshold concentrations of c-di-GMP and phenotypic output in the control of cell size and cell survival

The availability of dozens of mutants constructed during this study and the measurement of their intracellular c-di-GMP levels provided an opportunity to quantify the relationship between the c-di-GMP concentrations and the phenotype observed. We plotted the cell length and width of the WT and all mutant strains exhibiting varying c-di-GMP levels (Fig 5A). For comparative purposes, the c-di-GMP level of the WT was normalized to 1. The data revealed an S-shaped dose-response relationship between c-di-GMP levels and both cell length and width (Fig 5A). Analysis of cell length, width, and c-di-GMP concentrations across different mutants, including the genetic consequence of different mutants, identified two distinct functional thresholds of the c-di-GMP signal (Fig 5A). First, a morphological threshold: when c-di-GMP levels fall below about ~80±5% of that of the WT, the cell length and cell width were significantly reduced (Fig 5A). Interestingly, in the range between 80±5% and 50±5%, the output reflected in the reduction of both cell length and cell width is proportional to the drop of c-di-GMP levels, suggesting that c-di-GMP is an important factor for cell size control in *Anabaena*. Second, a viability threshold: a reduction of 50%±5% in c-di-GMP levels relative to the WT resulted in complete loss of cell viability (Fig 5A). These findings establish c-di-GMP as a critical regulator of cell size and survival of *Anabaena*.

To further confirm the relationship between c-di-GMP levels and phenotypic outputs, we titrated c-di-GMP concentrations using the conditional cdG$^0$ strain and then examined the phenotypic consequences. As shown in Fig 5B, the result confirmed the S-shaped dose-response relationship between c-di-GMP concentrations and both cell length and width (Fig 5B), replicating the two functional thresholds observed in different mutant strains (Fig 5A). Notably, while the morphological (~80±5% of WT) and viability (~50±5% of WT) thresholds remained consistent, this conditional system revealed an additional phenotype: when c-di-GMP levels decreased below 50±5% of WT, cells first exhibited extreme cell size reduction before progressing to complete cell death, revealing a distinct lag phase progressing to lethal effect, which was not apparent in other mutants. These results provided new insights into the c-di-GMP regulatory functions.

## Cell fate determination by c-di-GMP is dependent on its ability to titrate CdgR

CdgR is a known c-di-GMP-specific receptor that regulates cell size in *Anabaena* [7]. Several point mutations in CdgR that abolish c-di-GMP-binding lead to cell death [7], suggesting that the increasing abundance of the c-di-GMP-free form of CdgR may account for the observed phenotype. To clarify the genetic basis of c-di-GMP-dependent regulation, we deleted *cdgR* in the *cdG$^0$* background. This mutation restored both cell size and cell viability to the WT levels (Fig 2G and 2H), indicating that CdgR mediates the effects of c-di-GMP on these traits in *Anabaena*. We next quantified the intracellular CdgR levels in the WT and mutants with varying cell sizes and c-di-GMP concentrations (Figs 5C and S8). In the WT strain, CdgR and c-di-GMP concentrations were 4.44±0.19 μmol/L and 4.84±0.51 μmol/L, respectively, yielding a CdgR/c-di-GMP ratio of 0.92±0.04 (S8 Fig). Since a CdgR dimer binds two molecules of c-di-GMP [7], little c-di-GMP-free form of CdgR is thus available in the WT strain under normal conditions. In the *cdG$^0$* strain, removal of inducers led to a decline in c-di-GMP (see Fig 2 for details) and a concomitant accumulation of CdgR (Fig 5C), increasing the CdgR/c-di-GMP ratio (Fig 5D). The mechanism underlying this CdgR accumulation upon c-di-GMP depletion remains to be determined, but may involve c-di-GMP-mediated repression of *cdgR* expression or inhibition of CdgR protein degradation. When c-di-GMP decreased to 55±11% of the WT, close to the viability threshold, at 48 h after the removal of inducers, the CdgR/c-di-GMP ratio increased to 3.5±0.46 (Fig 5D). By 72 h, c-di-GMP is nearly depleted, with CdgR existing almost entirely in the c-di-GMP-free form (Fig 5D). Similar results were observed in selected mutants that contain different levels of c-di-GMP and a correspondingly reduced cell size (Fig 5C and 5D). By cross-examining all the available data (Figs 2 and 5), we found that the ratio of CdgR/c-di-GMP is about 1.43±0.16 when the c-di-GMP level is at the morphological threshold, and it is about 3.5±0.46 when the c-di-GMP level reaches the viability threshold. Thus, the ratios of CdgR to c-di-GMP act as a central

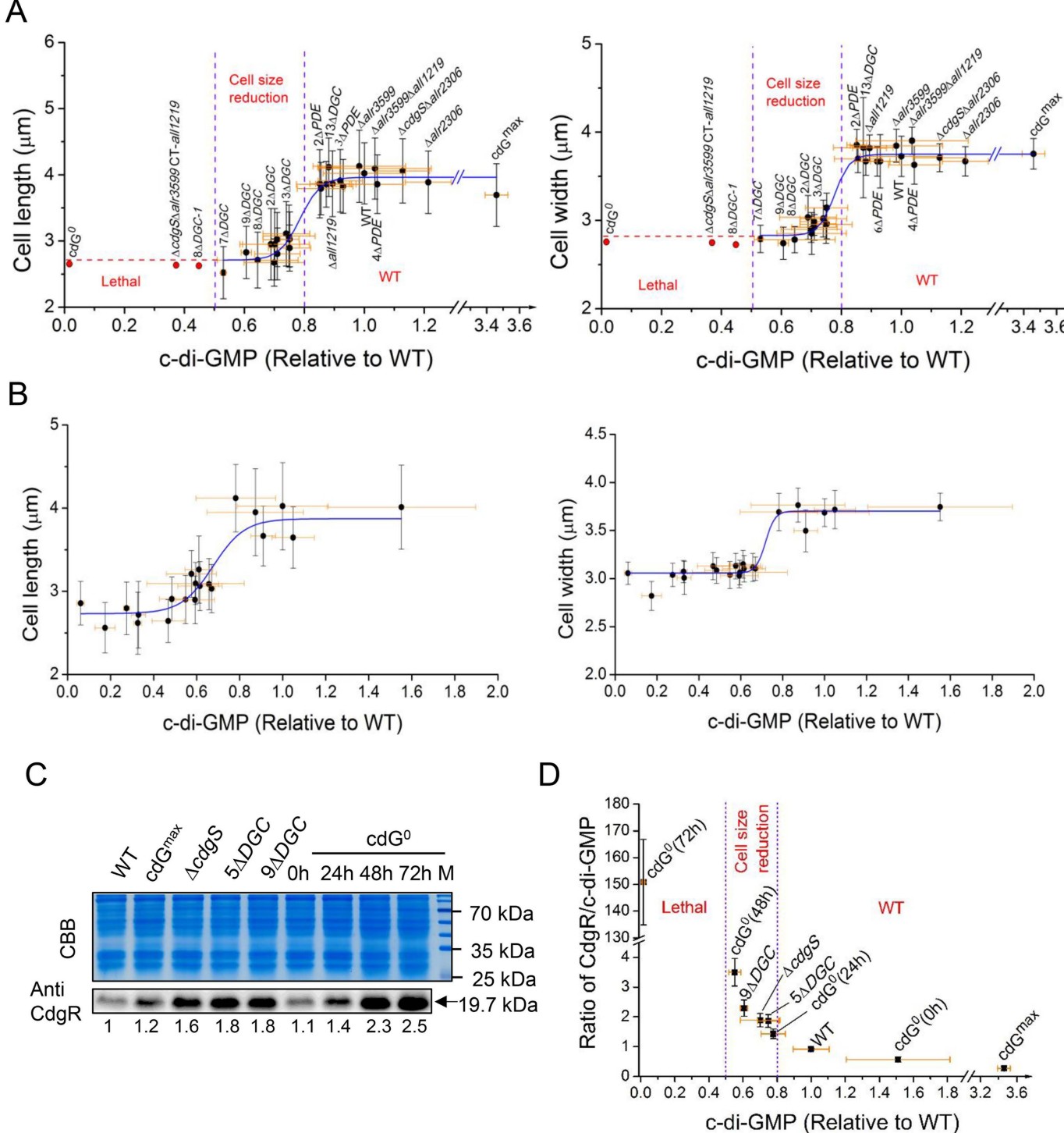

**Fig 5. The Interplay among c-di-GMP, CdgR, and cellular morphology in *Anabaena*.** (A) Correlation between intracellular c-di-GMP levels and cell length (left) and cell width (right) across different mutant strains. The red dot and dashed line indicate the lethal phenotype of the corresponding strains. Two concentration-dependent regulatory thresholds are indicated by purple dashed lines. (B) Dose-dependent effects of intracellular c-di-GMP levels on cell parameters (cell length (left) and width (right)) in the *cdG⁰* strain under varying concentrations of inducers (Cu²⁺ and theophylline). (C) CdgR abundance in the indicated variant strains and the cdG⁰ strain during the time course of Cu²⁺ and theophylline depletion. Total proteins were visualized with Coomassie Brilliant Blue (CBB), and CdgR was probed with a polyclonal antibody against CdgR (Anti-CdgR). CdgR levels corresponding to those

of the WT were quantified with ImageJ from three biological replicates. **(D)** Correlation between the molar ratios of CdgR to c-di-GMP calculated for the indicated strains and intracellular c-di-GMP levels. In (A) (B) (D), data points represent the mean±SD from biological replicates. The data underlying this Figure can be found in S1 Data. The raw images underlying this Figure can be found in S1 Raw Images.

determinant of the phenotypic output through a titration mechanism. These results support our previous hypothesis that the accumulation of the c-di-GMP-free form of CdgR ultimately determines cell fate in *Anabaena* [7].

## Discussion

Most bacterial genomes encode numerous enzymes for c-di-GMP synthesis and degradation [16,20,23]. It is also known that cyanobacteria modulate the c-di-GMP pool in a dynamic manner [4,5,16]. However, a systemic understanding of how the entire enzyme system coordinately regulates c-di-GMP homeostasis remains lacking in cyanobacteria. *Anabaena* possesses 16 c-di-GMP metabolic enzymes and has been reported to utilize c-di-GMP as a central regulator for cell size control [24,26]. In this study, we created many mutants, including two ($cdG^{max}$ and $cdG^0$) in which all genes encoding c-di-GMP synthesis and degradation enzymes were deleted, respectively (Figs 2 and S1–S4). Due to the relatively slow growth rate of cyanobacteria such as *Anabaena*, the deletion of such a large number of genes in one single mutant was unreported in cyanobacterial genetics. This long-term genetic effort led us to make several important conclusions on the function of the c-di-GMP signal in *Anabaena*.

First, our data indicate that c-di-GMP serves not only as a modulator for cell size control but also as an essential signaling molecule required for cell viability in *Anabaena* (Figs 2 and S1–S4). Second, quantitative analysis revealed two physiologically relevant c-di-GMP thresholds in vivo: a minimal level necessary for cell size maintenance and a critical lower limit required for cell viability (Fig 5). Third, we functionally classified the 16 c-di-GMP turnover enzymes into different categories based on their contribution to the global pool of c-di-GMP (Fig 6). Among them, the previously identified CdgS [26], All1219, and Alr3599 act as the primary DGCs (Figs 4 and S6), while Alr2306 serves as the primary PDE (Fig 3). These enzymes form a core enzymatic network that orchestrates c-di-GMP homeostasis to regulate cell size and viability (Fig 6). It also revealed a safeguard mechanism with Alr3599, which is induced post-transcriptionally to rescue the level of c-di-GMP when it drops below the viability threshold (Fig 4). Our results also demonstrate that both the cell size control by, and the essential function of, c-di-GMP are dependent on its ability to saturate CdgR, corresponding to the two threshold concentrations, and thus the two ratios of CdgR/c-di-GMP (Fig 5). Part of the functions of c-di-GMP-CdgR signaling pathway is mediated by DevH, an essential transcription factor in *Anabaena* [7,27]. Both DevH and NtcA are transcription factors belonging to the CRP family. The DevH regulon strongly overlaps that of NtcA and controls a multitude of cell physiology [27]. Other downstream factors, in addition to DevH, are likely involved in the control of cell size and viability.

Based on these findings, we propose an integrated regulatory model for c-di-GMP homeostasis in *Anabaena* (Fig 6). The 16 c-di-GMP metabolic enzymes are classified into three functional modules, according to their contribution to the c-di-GMP global pool (Fig 6). The basal module, comprising 13 enzymes including the dominant PDE Alr2306, maintains the basal c-di-GMP pool (~40% under our experimental conditions) (Fig 6). The primary module, which includes the major DGCs CdgS and All1219, contributes to around 60% of cellular c-di-GMP under our experimental conditions (Figs 6 and 4B). CdgS and CdgK constitute a two-component system, and CdgK possesses multiple sensor domains at the N-terminal [26]. Alr1219 also has CHASE and PAS sensor domains (Fig 1A). These observations suggest that the contribution of the primary module may further vary according to environmental or internal cues. Together with the basal c-di-GMP pool, they govern the cell fate of *Anabaena* through two c-di-GMP concentration thresholds: a morphological threshold (~80%) and a viability threshold (~50%). When total c-di-GMP (Basal+Primary modules) exceeds the morphological threshold, cell size is properly maintained. At intermediate levels (between the two thresholds), cell size reduces proportionally. If the levels of c-di-GMP fall below the viability threshold, loss of cell viability occurs. The different c-di-GMP threshold concentrations translate into corresponding c-di-GMP/CdgR ratios, which ultimately dictate cell fates and responses (Figs 5 and 6).

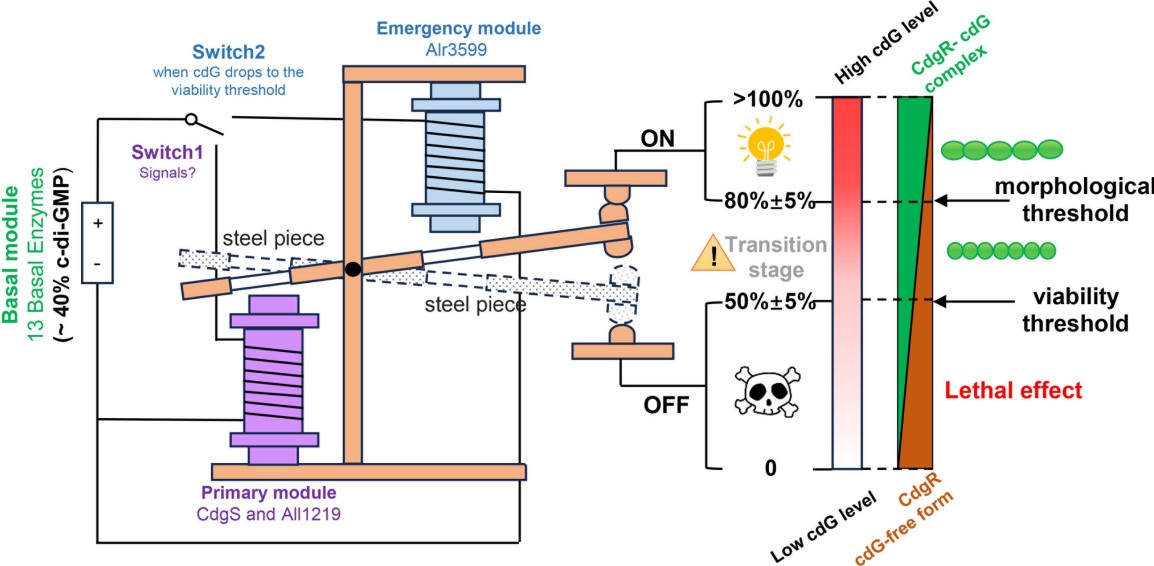

**Fig 6. A dual-threshold c-di-GMP regulatory system governs cell growth and survival in *Anabaena*.** Under our experimental conditions, the c-di-GMP levels are regulated by three enzymatic modules: (1) Basal module: maintains basal c-di-GMP pool (~40%) through the activity of 13 enzymes, including the dominant PDE Alr2306. This module is analogous to a battery in an electromechanical relay. (2) Primary module: comprises the DGCs CdgS and All1219, which operate in a manner like an electromagnetic coil. Their signal-dependent contribution (~60% of total c-di-GMP under our test conditions), together with the basal c-di-GMP pool, defines three phenotypic states: >80±5% c-di-GMP: normal *Anabaena* morphology (steel piece engaged); 50±5–80±5% c-di-GMP: cell size decreases proportionally with c-di-GMP level; <50±5 c-di-GMP: lethal effect in *Anabaena* (steel piece disengaged). (3) Emergency module: the dormant DGC Alr3599 activates when c-di-GMP drops to the viability threshold (50±5% c-di-GMP), functioning as a backup coil to save cells from death. A dual-threshold control mechanism (morphological threshold: 80±5%; viability threshold: 50±5%) ensures precise regulation of cellular c-di-GMP homeostasis. The phenotypic outcomes, including reduced cell size and lethality, are ultimately determined by the molar ratio of CdgR to c-di-GMP. Arrows indicate threshold values. *cdG*: c-di-GMP.

Certain mechanistic details within this model remain to be elucidated. These included the identification of the specific signals perceived by the sensor domains of CdgK-CdgS and Alr1219, as well as the precise post-transcriptional mechanism that activates Alr3599 under c-di-GMP-limited conditions. Moreover, given that all c-di-GMP metabolic enzymes possess putative sensor domains themselves (Fig 1A) and are thus likely responsive to distinct signals, their relative contributions to the global c-di-GMP pool, and consequently their functional categorization, may be dynamically reconfigured under different physiological conditions. Therefore, variations of the proposed model depicted in Fig 6 may exist. An important future direction is to investigate the plasticity of cell size in wild-type (WT) *Anabaena* in response to environmental changes and to determine if such plasticity is governed by dynamic fluctuations in the global c-di-GMP pool. It is noteworthy that the observed cell size reduction resembles certain physiological processes observed in related filamentous cyanobacteria, such as hormogonia formation or programmed cell death [31–33], suggesting that the c-di-GMP signaling pathway may participate in these processes. Finally, while our study establishes the essential roles of c-di-GMP in regulating cell size and viability, it is highly probable that c-di-GMP regulates additional cellular processes in *Anabaena*. Long-term laboratory cultivation of this strain may have led to the loss of certain phenotypes, such as motility, biofilm formation, and hormogonia formation [31,34]. However, since certain core components of these pathways are still conserved in *Anabaena* [35], they might still be regulated by c-di-GMP levels. Along the same line, the signaling mechanisms of c-di-GMP, revealed in *Anabaena*, offer a framework for the understanding of these important physiological processes in cyanobacterial strains in which these traits are still preserved. Exploring these potential connections will be crucial for a complete understanding of the functions of c-di-GMP in cyanobacterial physiology. Despite these open questions, the core

regulatory logic we propose here by integrating maintenance of basal c-di-GMP pool, signal-responsive modulation, and an emergency override system, provides a robust and precise framework for understanding c-di-GMP homeostasis not only in *Anabaena,* but also in other cyanobacteria.

In bacteria, multiple signaling pathways can operate concurrently to generate distinct physiological outputs while utilizing the same diffusible second messenger maintained at a consistent global concentration [36,37]. This remarkable signaling specificity arises from the integration of local and global c-di-GMP signaling modalities within complex regulatory networks [36]. Our study reveals that c-di-GMP regulates the cell size and viability in *Anabaena* (Fig 2). Unlike well-characterized systems where c-di-GMP mediates different cellular processes through local c-di-GMP signaling [36,38–41] (e.g., the c-di-GMP-dependent Bcs exopolysaccharide-producing system [38,41] and the Nfr vicinity system in *E. coli* [39,40]), our data suggest that *Anabaena* primarily employs global c-di-GMP signaling to coordinate cell size and viability (Fig 2C and 2H). Several lines of evidence support this conclusion. First, while targeted deletion of the diguanylate cyclase CdgS produces a specific phenotype in cell size reduction, the accompanying substantial changes in total cellular c-di-GMP levels strongly implicate a global signaling mechanism (Fig 2C). Second, both cell size and viability regulation converge on the same c-di-GMP effector, CdgR (Fig 2H), which shows no physical interaction with pathway-specific DGCs (S9 Fig). Notably, *Anabaena* also exhibits a distinct c-di-GMP regulatory logic compared to canonical paradigms. In most bacteria, elevated c-di-GMP levels promote sessile behaviors [42–44] (e.g., biofilm formation in *Pseudomonas aeruginosa* [45]). However, in *Anabaena*, elevated c-di-GMP levels did not induce phenotypic changes at the tested conditions, whereas c-di-GMP reduction triggered an adaptive response in cell size control (Fig 2) [26]. This inverse correlation suggests a fundamentally different wiring of c-di-GMP signaling networks in this organism.

Cell size homeostasis represents a fundamental biological characteristic across bacterial species [46,47]. Recent work has established ppGpp as a central regulator of bacterial cell size, directly linking metabolic status to division control through growth rate modulation and adder-based mechanisms in *E. coli* [48]. Strikingly, our studies reveal that *Anabaena* employs c-di-GMP, not ppGpp, as the primary determinant of cell size, while retaining functional ppGpp synthetases/hydrolases (S10 Fig). Overexpressing or disrupting the single relA/spoT homolog protein All1549 in *Anabaena* (strain OE-*all1549* or *all1549Ωsp/sm*) failed to alter cell size significantly (S10 Fig) [49], while perturbations in c-di-GMP pools elicited pronounced cell size effects (Fig 2). The observed divergence in cell size regulatory systems likely stems from unique physiological constraints imposed by the photosynthetic lifestyle and multicellular complexity of *Anabaena*. Except for the role of the DevH that accounts only partly for c-di-GMP signaling output [7,27], the downstream molecular mechanisms by which this second messenger governs cell size remain to be fully elucidated and represent an important direction for future investigation.

In summary, this study demonstrates how nature repurposes the conserved signaling molecule c-di-GMP to construct novel regulatory architectures adapted to specific environmental contexts. Beyond revealing novel adaptation strategies in cyanobacteria, this mechanism provides a blueprint for engineering synthetic biological systems that require multi-level regulation.

## Materials and methods

### Bacterial strains, media, and growth conditions

*Anabaena* sp. PCC 7120 and its derivatives were cultivated at 30 °C in BG11 medium under continuous illumination (white light, 30 µmol m⁻²·s⁻¹) with shaking at 180 rpm [50]. For conditional mutants, where gene expression was controlled by the artificial CT promoter [51], cells were initially grown in BG11 medium supplemented with 0.3 µM copper and 0.5~1.0 mM theophylline (TP) to induce gene expression during the logarithmic phase. To terminate gene expression, cells were washed three times with a BG11 medium devoid of copper and theophylline and then transferred to a fresh copper/TP-free BG11 medium. When required, Neomycin (25 µg/ml), streptomycin (2.5 µg/ml), or spectinomycin (5 µg/ml) was provided in the culture. All strains used in this study are listed in S1 Table.

## Construction of plasmids and cyanobacterial recombinant strains

All markerless deletion mutants, conditional mutants, and strains with point mutation were created using the genome editing technique based on CRISPR-Cpf1 as previously described [29,51,52]. Plasmids containing the artificial CT promoter for inducible gene expression (using copper and theophylline) were constructed based on the PCT vector. For protein expression in *E. coli*, the corresponding DNA fragments were amplified and sub-cloned into the pHTwinStrep vector containing two carboxyl-terminal Strep-tags at the C-terminal. Plasmids containing genes with point mutations were generated using quick-change mutagenesis, with the plasmid harboring the corresponding WT gene as the template.

Markerless deletion and overexpression strains were created by conjugating the corresponding plasmids into *Anabaena* [7,53]. For the *all1219* conditional mutant, the native promoter region of *all1219* was replaced with the artificial CT promoter. Point mutation strains were constructed as previously described [26]. For multiple gene deletion strains, the target genes were deleted step by step. Briefly, after deleting one gene, the mutation was verified to be fully segregated before proceeding to the next gene deletion. Each newly constructed mutant was checked to ensure that previously deleted genes remained fully segregated (S1 and S4 Figs). All primers and plasmids used in this work are listed in S2 and S3 Tables, respectively.

## Estimation of Intracellular Concentration of CdgR and c-di-GMP

The packed cell volume was determined as previously described with minor modifications [7]. Briefly, 9 mL of WT cell suspension ($OD_{750}$ = 0.462) was added to a hematocrit tube, and the sample was centrifuged in a hematocrit tube at $5,000 \times g$ for 30 min, yielding a packed cell volume of 10.75 μL. This value corresponds to a normalized packed cell volume of 2.58 μL/$OD_{750}$, which is consistent with results reported by Laurent and colleagues. The cell volume was also estimated geometrically by modeling the cell as a cylinder, using the formula: V = π × (1/2 cell width) × (1/2 cell width) × cell length. To quantify the cellular abundance of CdgR, an immunoblot assay was employed. Cell extracts normalized to 0.3 $OD_{750}$ units were resolved by SDS-PAGE alongside a standard curve generated from a dilution series of purified CdgR protein of known concentration. Following transfer, the membrane was probed with specific anti-CdgR antibodies. The band intensities from the cell lysates were interpolated against the standard curve to determine the absolute amount of CdgR. The concentration of CdgR and c-di-GMP was then calculated based on their quantification and the corresponding cell volume and cell numbers, with the final value representing the mean of three independent biological replicates.

## Protein expression and purification

*E. coli* BL21(DE3) cells harboring the pHTwinStrep-derived plasmids were cultured in LB medium supplemented with 50 μg ml$^{-1}$ kanamycin at 37 °C until reaching an $OD_{600}$ of 0.5–0.8. Protein expression was induced by adding 0.5 mM isopropyl-β-D-1-thiogalactopyranoside (IPTG) and incubating overnight at 16 °C. Cells were collected by centrifugation at 8,000 g for 5 min, resuspended in lysis buffer (50 mM Tris-HCl, pH 8.0, 0.5 M NaCl), and lysed using a French press. Cell debris was removed by centrifuging at 12,000 g at 4 °C for 35 min. The supernatant containing the soluble proteins was collected and mixed with preequilibrated Strep-Tactin XT Superflow resin (IBA) for 1 h at 4 °C. Then, the resin was collected by filtration and extensively washed with buffer W (50 mM Tris-HCl, pH 8.0, 0.2 M NaCl). The proteins were subsequently eluted with elution buffer (buffer W containing 50 mM biotin), concentrated (Amicon Ultracel-10K, Millipore), and further purified by size-exclusion chromatography (SEC) on a Superdex 200 Increase 10/300 GL column (GE Healthcare) equilibrated with buffer W. Fractions containing the recombinant protein were pooled, concentrated, and stored at −80 °C.

## Extraction and quantification of cellular c-di-GMP

At the indicated time points or the indicated conditions, 3 OD of cells were quickly collected by filtering and resuspended with 600 μL ice-cold extraction buffer containing 40% acetonitrile, 40% methanol, and 20% $H_2O$. The cells were broken by heating (95 °C for 15 min), followed by incubating at −20 °C for 30 min and then centrifuging at 12,000 g for 5 min at 4 °C.

The supernatant was transferred to a 2 ml tube and stored at −20 °C until further analysis. All experiments were repeated three times. For quantification, the supernatants were evaporated and dried by a vacuum manifold. The pellets were resuspended with 100 μL $H_2O$. 10 μL of each sample was analyzed by UPLC-MS/MS on an ACQUITY UPLC H-class Xevo TQ MS system, equipped with a Waters BEH C18 2.1×50 mm column. All samples were filtered with a 0.22 μm Ultra free-MC membrane before being subjected to the column. The standard curve of GTP or c-di-GMP (Biolog) with the concentrations of 400, 200, 100, 50, 25, 12.5, and 6.75 nM was used to calculate the GTP or c-di-GMP concentration of each sample, respectively.

### In vitro assays of DGC activity

For DGC activity, 1 μM protein was incubated with 25 μM GTP (Thermo Scientific) in reaction buffer (50 mM Tris-HCl, pH 8.0, 0.2 M NaCl, 10 mM $MgCl_2$). After incubation for 10 min at 25 °C, the reactions were quenched by the addition of 2 μL EDTA (0.5 M), followed by heating up to 95 °C for 15 min. After heating, the samples were analyzed by HPLC.

For the kinetic parameter of DGC activity, a series of concentrations of GTP (3.13, 6.25, 12.5, 25, 50, 100, and 200 μM, respectively) was incubated with 1 μM protein for 10 min at 25 °C. Then all reactions were quenched and analyzed by HPLC as described above for DGC activity. For inhibition assays, the protein was preincubated with varying concentrations of c-di-GMP (1–100 μM) for 5 min at 25 °C, followed by the addition of 25 μM GTP. The reaction was stopped at indicated time intervals by adding 2 μL EDTA (0.5M). The kinetic parameters of DGC activity and inhibition constants were determined using GTP consumption according to the corresponding substrate concentration as previously described [26].

### Quantitative real-time PCR (qRT-PCR) and western blots

The Δ*cdgSCT-all1219* strain was cultivated to the log phase in BG11 with 0.3 μmol $Cu^{2+}$ and 1 mM theophylline, and then washed with BG11 without inducer three times. Cells were resuspended in 200-ml fresh BG11 with or without inducer to an $OD_{750}$ of 0.3 and cultured under the same conditions for 4 days. Ten OD of each sample was harvested at 0 h, 24 h, 48 h, 72 h, and 96 h, respectively, with three parallel experiments. Total RNA was extracted with Plant Total RNA Isolation Kit (FOREGENE) and reverse transcribed with HiScript Q RT super mix (Vazyme Biotech Co., Ltd, Nanjing, China). qRT-PCR was performed by C1000 Touch Thermal Cycler (Bio-Rad Laboratories, Hercules, CA, USA) using ChamQ SYBR qPCR Master Mix (Vazyme Biotech Co., Ltd) with specific primers listed in S2 Table.

The Δ*cdgSCT-alr1219 (alr3599-Flag)* cell samples were prepared as above to determine the Alr3599 protein expression level by western blot analysis. The collected cells were resuspended in protein loading buffer, then broken using a FastPrep-24 (6.0 m/s; QuickPrep, 60 s), and subsequently boiled for 10 min. After centrifuging at 13,500 rpm for 10 min, the supernatant was separated on 10% SDS-PAGE gels by standard electrophoresis and transferred to PVDF membranes. Blots were probed with a Flag (C-Term) antibody (Sigma- Aldrich, F1804, 1:5000) and HRP-conjugated secondary antibody (1:5000, goat anti-mouse).

### Image acquisition and cell size analysis

The microscope images were captured by the SDPTOP EX30 microscope and processed using ImageJ without deconvolution as described previously [51]. For cell size analysis, microscopy images were taken when the optical density of a culture reached 0.3–0.5 at $OD_{750}$. The cell length and width of cells of each strain were measured using ImageJ. Statistical tests and plotting of data were performed with Origin 8.0.

### Statistics and reproducibility

All statistical analyses and data visualization were performed using SPSS software version 19.0 (SPSS) and Origin software, respectively. Each data point in the plots represents an independent measurement derived from at least two or three biologically independent experiments. Comparison between the control and tested groups was performed using an

independent sample $t$ test. Multiple comparisons were done using One-Way ANOVA followed by Sidak's multiple comparisons test. Significance thresholds ($p$-values) are provided in the figure legends.

## Supporting information

**S1 Fig. PCR verification of the *cdG^max* strain.** (**A**) A schematic diagram illustrating the primer design strategy for PCR verification. (**B**)Verification of the deletion of 8 genes in the *cdG^max* strain via PCR. X: *cdG^max*, C: WT control. M: DNA marker. The raw images underlying this Figure can be found in S1 Raw Images.
(DOCX)

**S2 Fig. Micrographs of filaments of related PDEs deletion mutant strains.** Micrographs of filaments of related PDEs deletion mutant strains. Micrographs of *Anabaena* filaments of the indicated strain. WT: wild-type. Scale bars: 15 µm. The raw images underlying this Figure can be found in S1 Raw Images.
(DOCX)

**S3 Fig. Growth curves of WT and *cdG^max*.** The initial concentration, measured at OD750 for each strain, was 0.025. The strains were then cultivated in BG11 medium or BG11$_0$ medium at the indicated time. All values are shown as mean ± standard deviation, calculated from triplicate data. The data underlying this Figure can be found in S1 Data.
(DOCX)

**S4 Fig. Verification of a conditional mutant of the *cdG^0* strain (14Δ*DGC*).** (**A**) A schematic diagram and PCR verification of the conditional control of all1219 in the cdG0 strain. Left panel: the cdG0 strain (*14ΔDGC*) is a conditional mutant. The promoter region of all1219 was replaced by a $Cu^{2+}$ and theophylline (TP) inducible platform (CT promoter) at the native chromosomal locus. Right panel, PCR verification of conditional mutant. M: DNA marker. (**B**) Verification of the deletion of the other 13 genes in the *cdG^0* strain by PCR. Primers for each gene were designed as depicted in Fig S1A. X: *cdG^0*, C: WT control. M: DNA marker (**C**) Cultures of the cdG0 strain and WT strain in BG11 medium with or without $Cu^{2+}$ and TP, which were photographed after 8 days (top) and 20 days (bottom) of incubation. The raw images underlying this Figure can be found in S1 Raw Images.
(DOCX)

**S5 Fig. Micrographs of *Anabaena* filaments of WT and different deletion strains.** Micrographs of *Anabaena* filaments of the indicated strain. Scale bars: 15 µm. WT, wild-type *Anabaena*. The raw images underlying this Figure can be found in S1 Raw Images.
(DOCX)

**S6 Fig. The enzymatic characterization of CdgS, All1219-ΔCT, and Alr3599 in vitro.** (**A**) The DGC activity of CdgS, All1219-ΔCT, and Alr3599 was evaluated by the reaction product of c-di-GMP. All reaction products are assessed by the retention time of HPLC and correspond to standards. The top panel shows the retention time of the standard nucleotides, including c-di-GMP and GTP. (**B**) Analysis of the kinetics of c-di-GMP synthesis activity of CdgS, All1219-ΔCT, and Alr3599. The down panel shows curves fitting to the Michaelis–Menten equation by nonlinear regression analysis, and the upper panel shows the enzyme kinetic parameters. Assays were performed using GTP as substrate at varying concentrations. The assays were performed three times, and the average value and standard deviations are shown. (**C**) Determination of Ki of CdgS, All1219-ΔCT, and Alr3599 for c-di-GMP. Inhibition of the specific activity of CdgS, All1219-ΔCT, and Alr3599 over time was measured in the presence of different c-di-GMP concentrations. Points and error bars represent the mean ± standard deviation (SD) calculated from three biological replicates. The data were fitted to an inhibition model with a variable slope. (**D**) qRT-PCR analysis of gene *alr3599* in the Δ*cdgSCT-all1219* strain cultured in BG11 medium with (+) or without (−) $Cu^{2+}$ and theophylline at the indicated time points. Transcript levels are shown as mean ± SD from three

biological replicates. (**E**) Western blotting analysis of Alr3599 protein levels in the Δ*cdgSCT-all1219(alr3599-Flag)* strain cultured in BG11 medium with (+) or without (−) $Cu^{2+}$ and theophylline at the indicated time points. Similar amounts of total proteins extracted from different samples were loaded on the gel, stained with Coomassie Brilliant Blue (Top, CBB), or probed with a polyclonal antibody against Flag (Bottom). M: maker. The data underlying this Figure can be found in S1 Data. The raw images underlying this Figure can be found in S1 Raw Images.
(DOCX)

**S7 Fig. Micrographs of WT and Δ*cdgS* pICT-*all1219* (*alr3599*-Flag) strains.** Micrographs of *Anabaena* filaments of WT and Δ*cdgS* pICT-*all1219* (*alr3599*-Flag) strains cultured at the indicated times and conditions (with or without inducer ($Cu^{2+}$ and theophylline (TP)) in BG11 medium). Scale bars represent 15 µm. The raw images underlying this Figure can be found in S1 Raw Images.
(DOCX)

**S8 Fig. Quantification of CdgR and c-di-GMP concentrations in *Anabaena*.** (Upper): representative immunoblot for the quantification of CdgR molecules. Cell lysates derived from a culture normalized to $OD_{750} = 0.3$ were analyzed alongside a standard curve of purified CdgR (shown on the left) via immunoblotting with anti-CdgR antibodies. Total proteins were loaded and stained with Coomassie Brilliant Blue (CBB). (Lower) The calculated average intracellular concentrations of CdgR and c-di-GMP in WT *Anabaena* cells, derived from the immunoblot, c-di-GMP quantification, and packed cell volume measurements. The raw images underlying this Figure can be found in S1 Raw Images.
(DOCX)

**S9 Fig. Major DGCs are not interacting with the c-di-GMP receptor (CdgR).** Bacterial two-hybrid (BACTH) assay for the interaction of the indicated proteins. BTH101 cells producing the indicated proteins fused to either the T18 or the T25 domain of the adenylate cyclase were spotted on plates supplemented with IPTG and X-Gal. The gene encoding the leucine zipper region of GCN4 protein (ZIP) and the empty vectors (EV) were used as positive and negative controls, respectively. Interaction between the two fusion proteins is attested by the blue color of the colony. The experiment was conducted three times, and one representative plate is shown. The raw images underlying this Figure can be found in S1 Raw Images.
(DOCX)

**S10 Fig. Overexpression or disruption of All1549 (RelA/SpoT homolog) does not alter cell morphology in *Anabaena*.** (**A**) Left panel: micrographs of *Anabaena* filaments of WT and OE- *all1549* strains cultured with inducer (1 µM $Cu^{2+}$ and 2 mM theophylline in BG11 medium). Scale bars represent 15 µm. Right panel: statistical analysis of cell length and cell width of the indicated strains based on images, as shown in the right panel, using a box plot. 200 cells of each strain from three independent experiments were measured. (**B**) Left panel: micrographs of *Anabaena* filaments of WT and *all1549Ωsp/sm* strains cultured in BG11 medium. Scale bars represent 15 µm. Right panel: statistical analysis of cell length and cell width of the indicated strains based on images, as shown in the right panel, using a box plot. 200 cells of each strain from three independent experiments were measured. The data underlying this Figure can be found in S1 Data. The raw images underlying this Figure can be found in S1 Raw Images.
(DOCX)

**S1 Table. Strains used in this study.**
(DOCX)

**S2 Table. Primers used in this study (sequence in minuscule corresponds to the overlapping homologous parts in PCR fragments for ligation during cloning, and the sequence in capital letters corresponds to those used for DNA amplification by PCR).**
(DOCX)

**S3 Table. Plasmids used in this study.**
(DOCX)

**S1 Data. Underlying numerical data for figures in this study.** This file contains the individual numerical values underlying the summary data presented in Figs 2A, 2B, 2C, 2D, 2F, 2G, 3A–3D, 4A, 4B, 4F, 5A, 5B, 5D, S3, S6, S8, and S10. Each sheet corresponds to a specific figure panel. Mean values and error bars were calculated from the individual biological replicates as indicated.
(XLSX)

**S1 Raw Images. Original, uncropped images supporting all micrographs and blot results are presented in the main figures and supplemental figures.**
(PDF)

## Acknowledgments

We want to thank Jun Men and Siyu Wang from the Analysis and Testing Center of the Institute of Hydrobiology, Chinese Academy of Sciences, for their support with the HPLC and LC-MS/MS measurements.

## Author contributions

**Conceptualization:** Xiaoli Zeng, Cheng-Cai Zhang.

**Formal analysis:** Qing-Xue Sun, Yiling Yang, Xiaoli Zeng.

**Funding acquisition:** Qing-Xue Sun, Xiaoli Zeng, Cheng-Cai Zhang.

**Investigation:** Qing-Xue Sun, Min Huang.

**Methodology:** Qing-Xue Sun.

**Visualization:** Qing-Xue Sun, Xiaoli Zeng.

**Writing – original draft:** Xiaoli Zeng.

**Writing – review & editing:** Xiaoli Zeng, Cheng-Cai Zhang.

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
