## [Editor Report · Decision Letter 0]

10 Dec 2025

Dear Dr Zhang,

Thank you for submitting your manuscript entitled "An Electromechanical Relay-like Multi-enzymatic System Gates c-di-GMP-Dependent Cell Fate in a Cyanobacterium" for consideration as a Research Article by PLOS Biology.

Your manuscript has now been evaluated by the PLOS Biology editorial staff as well as by an academic editor with relevant expertise and I am writing to let you know that we would like to send your submission out for external peer review. However, please note that due to the coming holidays, we would only be able to get back to you with a decision at the beginning of January.

Before we can send your manuscript to reviewers, we need you to complete your submission by providing the metadata that is required for full assessment. To this end, please login to Editorial Manager where you will find the paper in the 'Submissions Needing Revisions' folder on your homepage. Please click 'Revise Submission' from the Action Links and complete all additional questions in the submission questionnaire.

Once your full submission is complete, your paper will undergo a series of checks in preparation for peer review. After your manuscript has passed the checks it will be sent out for review. To provide the metadata for your submission, please Login to Editorial Manager (https://www.editorialmanager.com/pbiology) within two working days, i.e. by Dec 12 2025 11:59PM.

Kind regards,

Melissa

Melissa Vazquez Hernandez, Ph.D.

Associate Editor

PLOS Biology

---

## [Decision Letter · Decision Letter 1]

28 Jan 2026

Dear Cheng-Cai,

Thank you for your patience while your manuscript "An Electromechanical Relay-like Multi-enzymatic System Gates c-di-GMP-Dependent Cell Fate in a Cyanobacterium" was peer-reviewed at PLOS Biology. It has now been evaluated by the PLOS Biology editors, an Academic Editor with relevant expertise, and by two independent reviewers.

In light of the reviews, which you will find at the end of this email, we would like to invite you to revise the work to thoroughly address the reviewers' reports. As you will see below, the reviewers are positive about the relevance and novelty of the study, yet some concerns have raised during revision. Reviewer 1 raises concerns about conceptual framing, clarity of the main message, and presentation, particularly questioning the usefulness of the “electromechanical relay” model, asking for clearer articulation of the global c-di-GMP pool concept, better contextualization with wild-type physiology and prior cyanobacterial literature, and improved organization and methodological transparency. The reviewer does have a few concerns that may require additional experimental work like the possibility of other c-di-GMP-regulated processes beyond cell dimensions. Reviewer 2 raises an important conceptual concern that the manuscript may overstate cell size and viability as the sole c-di-GMP outputs in a domesticated laboratory strain, potentially overlooking roles in motility, biofilm formation, and hormogonium development. Together, the reviews suggest the study is strong but requires clearer conceptual framing, broader biological context, and targeted clarifications before further consideration. We agree with all reviewer concerns which MAY require some additional experimental revisions to address them.

**IMPORTANT - SUBMITTING YOUR REVISION**

*Re-submission Checklist*

*Published Peer Review*

*PLOS Data Policy*

*Blot and Gel Data Policy*

Sincerely,

Melissa

Melissa Vazquez Hernandez, Ph.D.

Associate Editor

PLOS Biology

REVIEWERS' COMMENTS

Reviewer #1 (Conrad Mullineaux):

In this study, Sun et al probe the regulation of level of the bacterial second messenger c-di-GMP in the filamentous cyanobacterium Anabaena by constructing various kinds of mutants in 16 genes encoding enzymes responsible for c-di-GMP synthesis and/or c-di-GMP breakdown. C-di-GMP levels are measured in the mutants, and phenotypic effects are assessed by measuring growth rates and cell dimensions. There is also kinetic analysis of the enzymatic activity of some of the proteins. The experimental work all appears to have been done to a very high standard, and this is a technical tour de force - the construction, for example, of a mutant with simultaneous deletion of 14 genes is unprecedented in Anabaena, which is slow-growing, multicellular and polyploid. The main challenge is then trying to make a coherent story out of such a mountain of data. Here there is room for improvement. The authors' large collection of mutants enabled them to titrate c-di-GMP levels vs phenotype, and show the range of c-di-GMP levels over which c-di-GMP influences cell dimensions (Fig 4). They could further show a threshold level of c-di-GMP above which it no longer has any further effect on cell dimensions, and a second threshold below which further depletion of c-di-GMP is lethal. This is very interesting, and together with the data on the heterologous expression of a DGC (Fig 1 G,H) it strongly suggests that the multiple c-di-GMP metabolising enzymes in Anabaena exert their effects by acting as independent inputs into a global c-di-GMP pool, in contrast to the more usual picture of highly specific and localised c-di-GMP effects. For me that is the main message of the work. The authors discuss it (lines 569-584) but overall I think they need to be a lot clearer on what their study shows and what remains to be understood. Specific points:

1. The authors make a big feature of their analogy with an electromechanical relay, which features in the title and takes up a whole figure (Fig. 5) and many lines of the discussion. It would be interesting to know what other reviewers made of it, but for me it didn't help at all - in fact I found it made the message harder to understand, as I had to try to understand a very complex and imperfect analogy in addition to trying to understand the system in Anabaena. I am not sure about lumping 13 enzymes together as a "basal power unit". One enzyme could do that job - isn't it more likely that all these enzymes are regulated differently to provide independent inputs into a global c-di-GMP pool? That would make this an interesting system for integrating information from multiple sensors. In any case, I would recommend that the authors find another way to present their model for the Anabaena system.

2. The discussion focuses on the phenotypes of mutants vs the wild type, but has nothing to say about regulation of cell size in the wild type. How plastic is cell size in wild type Anabaena and is there evidence for changes in global c-di-GMP levels in wild type Anabaena according to conditions? Is there any evidence for programmed cell death in Anabaena (as would be induced by very low c-di-GMP levels)? It is possible, as related species form necridia, for example to liberate hormogonia from the branched filament network in Fischerella/Mastigocladus species.

3. Construction of mutants. Anabaena is polyploid and so it important to check for homozygosity of null mutants, whether they are generated by CRISPR or by any other method. Authors from the same group make that point in ref 46. Here, I assume that each null mutation was checked for full segregation, but, in the paper, nothing is said about that - information about how this was done must be provided.

4. In their discussion the authors focus on cell dimensions, as that is what they look for, but it seems very likely that c-di-GMP controls other things too - I think that should be acknowledged in the discussion.

5. I found it impossible to keep track of the authors' arguments without constant reference to Table S1, which lists the genes involved and the mutants constructed. So that is essential information - it belongs in the main paper, not the supplementary. In fact it would be useful to add an extra column to Part A with comments on conclusions on regulation and function from this study.

6. Lines 470-471: "c-di-GMP pool dynamics remain poorly characterized, particularly in cyanobacteria". That ignores a substantial literature in unicellular cyanobacterial species, which deserves more discussion here. See eg ref 4.

7. Line 101 - should be "homeostasis"?

8. Line 263 - "maintaining" is confusing, as it suggests that Alr2306 is supplying c-di-GMP when in fact it seems to be acting mainly to break it down. "Controlling" would be better.

9. Line 244, Fig. 2 legend. "p < 0.000" ??

10. Lines 566-568: "Employing threshold-dependent state transitions resolves the fundamental growth survival trade-off that challenges most living systems" (and a similar statement at the end of the abstract). I don't understand what that means. Is there even a growth-survival trade-off in this situation?

11. Line 615. More details are needed on illumination, as light has been shown to have strong effects on c-di-GMP metabolism in cyanobacteria (eg ref 4).

12. Line 668. Missing reference.

Reviewer #2:

The manuscript by Sun et. al. describes a comprehensive study of the role of c-di-GMP related genes in the filamentous cyanobacterium Anabaena (Nostoc) 7120. Using a comprehensive genetic approach they sequentially inactivate all of the genes responsible for c-di-GMP synthesis or degradation, demonstrating that a minimal threshold of c-di-GMP is required for cell viability, and that a second, higher threshold acts as a switch regulating cell size. The authors also identify key genes involved in maintaining c-di-GMP homeostasis that ensure levels do not drop below the critical threshold for viability. Overall, this is an impressive study in its thorough, comprehensive genetic approach to exploring the role of c-di-GMP in Anabaena 7120. The manuscript is generally well written, with a few minor editorial concerns listed below, and the results largely support the authors conclusions. The only major concern I have with the manuscript is that it fails to address how the loss of motility and biofilm formation in this organism due to laboratory domestication might obscure key roles for several of these genes. As written, the authors seem to imply that cell size and viability are the only two phenotypes regulated by c-di-GMP in this filamentous cyanobacterium. While this might be true for the commonly used lab strain, it may not reflect the entirety of the role of c-di-GMP in natural isolates of this strain and other closely related filamentous cyanobacteria. Other studies have demonstrated key roles for c-di-GMP in motility and biofilm formation for other filamentous cyanobacteria and it would be worth discussing all of this. On a related note, have the authors considered the possibility that the link between c-di-GMP and cell size regulation may be related to activation of dormant hormogonium development? Hormogonium development is known to be accompanied by a rapid reduction in cell size and shape, and while 7120 has lost the ability to differentiate hormogonia, it is possible that part of this pathway is activated by the decreased c-di-GMP levels, which could explain the biological relevance of cell size control via c-di-GMP.

Fig. 1D - are the cdG0 strains mislabeled, strain with inducer (green line) fails to grow, but strain without inducer (pink line) does grow? Opposite of what is reported in text and inconsistent with other data.

Line 186-187 - should this be figure 1D? Fig. 1C measures c-di-GMP levels, not growth.

Line 212 - insert "a" between "in" and "detectable"

Line 276-277 - As the authors note, it is surprising that removal of inducer does not lead to lack of cell growth in the DcdgSCT-all1219 strain, given that a cdgS-all1219 double mutant could not be generated. Could the authors comment on why this might be the case?

Line 373 - insert "during" between "homeostasis" and "standard"

Line 427 - would "phenotypic consequences" be more appropriate than "genetic consequences" here?

442 - Is apo-form the appropriate term here? Apo-form typically refers to the inactive form of an enzyme, but here it seems that it is the form that does not have c-di-GMP bound that is the active form, leading to growth arrest, and that c-di-GMP is an inhibitor of cdgR

Line 453 - why does reduced c-di-GMP lead to increased levels of CdgR? Is the mechanism known? If not, could the authors speculate on what why this occurs here or in the discussion?

Line 577-591 - Given that laboratory isolates of this organism are not known to display motility or biofilm formation, it is perhaps not surprising that perturbing c-di-GMP levels would not influence these phenotypes. However, this is likely due to other

---

## [Decision Letter · Decision Letter 2]

20 Mar 2026

Dear Dr Zhang,

Thank you for your patience while we considered your revised manuscript "A Dual-Threshold System Relying on Multiple c-di-GMP Metabolic Enzymes Controls Cell Fate of a Cyanobacterium" for publication as a Research Article at PLOS Biology. This revised version of your manuscript has been evaluated by the PLOS Biology editors, the Academic Editor and the original reviewers.

Based on the reviews and on our Academic Editor's assessment of your revision, we are likely to accept this manuscript for publication, provided you satisfactorily address the following data and other policy-related requests:

* Please add the links to the funding agencies in the Financial Disclosure statement in the manuscript details.

* DATA POLICY:

Regardless of the method selected, please ensure that you provide the individual numerical values that underlie the summary data displayed in the following figure panels as they are essential for readers to assess your analysis and to reproduce it: 2ABCFG, 3ABCD, 4F, S6D and S10AB.

* CODE POLICY

Per journal policy, if you have generated any custom code during the course of this investigation, please make it available without restrictions. Please ensure that the code is sufficiently well documented and reusable, and that your Data Statement in the Editorial Manager submission system accurately describes where your code can be found. More information on our Code Policy, what and how to share can be found here: https://journals.plos.org/plosbiology/s/code-availability

* As some of your findings rely on the microscopy images, we would like to encourage you to submit all other microscopy pictures to Zenodo or FigShare so they can be available to the readers too.

* Please add a scale bar to the microscopy images.

* We do not have a word limit. Please move the supplementary methods to the main text which can provide the readers an easier access to all information.

* Please move all references to the main reference list, or at least ensure that they are listed there as well. The search engines to not pick up references from the SI files, so those authors would not be credited for their work if it is only referenced in the SI file.

We expect to receive your revised manuscript within two weeks.

*Published Peer Review History*

*Press*

Sincerely,

Christian

Christian Schnell, Ph.D.

Senior Editor

PLOS Biology

cschnell@plos.org

on behalf of

Melissa

Melissa Vazquez Hernandez, Ph.D.

Associate Editor

PLOS Biology

Reviewer #1 (Conrad Mullineaux signed his report): It's a very comprehensive response to the reviewers' comments on the first submission. I found the revised manuscript much clearer and have no further modifications to suggest.

Reviewer #2: The authors have sufficiently addressed the reviewer comments from the previous submission.

---

## [Editor Report · Decision Letter 3]

25 Mar 2026

Dear Dr Zhang,

Thank you for the submission of your revised Research Article "A Dual-Threshold System Relying on Multiple c-di-GMP Metabolic Enzymes Controls Cell Fate of a Cyanobacterium" for publication in PLOS Biology. On behalf of my colleagues and the Academic Editor, Erin Goley, I am pleased to say that we can in principle accept your manuscript for publication, provided you address any remaining formatting and reporting issues. These will be detailed in an email you should receive within 2-3 business days from our colleagues in the journal operations team; no action is required from you until then. Please note that we will not be able to formally accept your manuscript and schedule it for publication until you have completed any requested changes.

PRESS

We frequently collaborate with press offices. If your institution or institutions have a press office, please notify them about your upcoming paper at this point, to enable them to help maximize its impact. If the press office is planning to promote your findings, we would be grateful if they could coordinate with biologypress@plos.org. If you have previously opted in to the early version process, we ask that you notify us immediately of any press plans so that we may opt out on your behalf.

Sincerely,

Christian

Christian Schnell, Ph.D.

Senior Editor

PLOS Biology

cschnell@plos.org

on behalf of

Melissa Vazquez Hernandez, Ph.D.

Associate Editor

PLOS Biology
